# After Ischemic Stroke, Minocycline Promotes a Protective Response in Neurons via the RNA-Binding Protein HuR, with a Positive Impact on Motor Performance

**DOI:** 10.3390/ijms24119446

**Published:** 2023-05-29

**Authors:** Katarzyna Pawletko, Halina Jędrzejowska-Szypułka, Katarzyna Bogus, Alessia Pascale, Foroogh Fahmideh, Nicoletta Marchesi, Aniela Grajoszek, Daria Gendosz de Carrillo, Jarosław Jerzy Barski

**Affiliations:** 1Department of Physiology, Faculty of Medical Sciences in Katowice, Medical University of Silesia, Medyków 18, 40-752 Katowice, Poland; agrajoszek@sum.edu.pl (A.G.); dariagendosz@gmail.com (D.G.d.C.); jbarski@sum.edu.pl (J.J.B.); 2Department for Experimental Medicine, Medical University of Silesia, Medyków 4, 40-752 Katowice, Poland; 3Department of Histology, Faculty of Medical Sciences in Katowice, Medical University of Silesia, Medyków 18, 40-752 Katowice, Poland; kasiabogus@outlook.com; 4Department of Drug Sciences, Pharmacology Section, University of Pavia, Viale Taramelli 14, 27100 Pavia, Italy; alessia.pascale@unipv.it (A.P.); foroogh.ft@gmail.com (F.F.); nicoletta.marchesi@unipv.it (N.M.); 5Department of Histology and Cell Pathology, Faculty of Medical Sciences in Zabrze, Medical University of Silesia, Poniatowskiego 15, 40-055 Katowice, Poland

**Keywords:** ischemic stroke, penumbra, minocycline, HuR, RNA-binding protein, inflammation, motor test

## Abstract

Ischemic stroke is the most common cause of adult disability and one of the leading causes of death worldwide, with a serious socio-economic impact. In the present work, we used a new thromboembolic model, recently developed in our lab, to induce focal cerebral ischemic (FCI) stroke in rats without reperfusion. We analyzed selected proteins implicated in the inflammatory response (such as the RNA-binding protein HuR, TNFα, and HSP70) via immunohistochemistry and western blotting techniques. The main goal of the study was to evaluate the beneficial effects of a single administration of minocycline at a low dose (1 mg/kg intravenously administered 10 min after FCI) on the neurons localized in the penumbra area after an ischemic stroke. Furthermore, given the importance of understanding the crosstalk between molecular parameters and motor functions following FCI, motor tests were also performed, such as the Horizontal Runway Elevated test, CatWalk™ XT, and Grip Strength test. Our results indicate that a single administration of a low dose of minocycline increased the viability of neurons and reduced the neurodegeneration caused by ischemia, resulting in a significant reduction in the infarct volume. At the molecular level, minocycline resulted in a reduction in TNFα content coupled with an increase in the levels of both HSP70 and HuR proteins in the penumbra area. Considering that both HSP70 and TNF-α transcripts are targeted by HuR, the obtained results suggest that, following FCI, this RNA-binding protein promotes a protective response by shifting its binding towards HSP70 instead of TNF-α. Most importantly, motor tests showed that reduced inflammation in the brain damaged area after minocycline treatment directly translated into a better motor performance, which is a fundamental outcome when searching for new therapeutic options for clinical practice.

## 1. Introduction

Stroke is the most common cause of adult disability and one of the leading causes of death worldwide. Approximately 87% are ischemic [1,2,3,4,5], with a serious socio-economic impact [6]. The therapeutic actions aimed at saving the hypoxic area (the penumbra) must be balanced with the risks of negative consequences associated with the undertaken treatment. Moreover, a very important factor within the therapeutic approach, primarily based on thrombolysis or thrombectomy, is the narrow “time window” available. This drawback renders the use of the most effective therapies (i.e., alteplase) ineffective in protecting against secondary damage [7,8]. Therefore, due to these limitations, only a small percentage of patients (<10%) affected by ischemic stroke qualify for this type of intervention [9,10]. Hence, the search for new therapeutic approaches for patients affected by ischemic stroke is a compelling medical need [11,12]. Within this context, the use of animal models is one of the most valid strategies to develop effective therapies [11,13]. Notably, the animal model of focal cerebral ischemia (FCI) allows for more reproducible results compared to other models [14,15,16] and closely mimics what happens in humans (i.e., the ischemic area takes a relatively small area of the entire cerebral cortex) [17,18]. Furthermore, this model is characterized by a low burden on the animals and enables the verification of the tested substance’s effect through motor tests [17,19]. A poor prognosis in ischemic stroke results from the irreversible loss of neurons [20], which is due to an abrupt blood flow shortage and excitotoxicity. Inflammation persists over an extended period, beginning at the time of stroke. Consequently, an increased level of pro-inflammatory cytokines and chemokines is observed, together with the infiltration of leukocytes into the areas affected by ischemia. Several studies have reported that pro-inflammatory cytokines (i.e., interleukin-1, interleukin-6 and TNFα (tumor necrosis factor alpha)) play a major role in the development of inflammation after stroke [21,22,23,24]. This suggests that anti-inflammatory treatments could expand the “therapeutic window” and facilitate the implementation of current clinical interventions [25]. Moreover, post-stroke inflammation plays an important role in the survival and regeneration of nerve cells, exerting a beneficial effect on brain tissue [26,27]. Therefore, in the search for new pharmacological strategies, the modulation of inflammation has been identified as a promising therapeutic option [28]. Antibiotics, especially tetracyclines, have gained a growing interest in this regard. Minocycline is a broad-spectrum antibiotic belonging to the tetracycline group with anti-inflammatory, antioxidant, and anti-apoptotic effects [29,30,31,32]. Previous studies on stroke have shown that the administration of minocycline attenuates neurological deficits and reduces ischemic infarct volume [8]. The anti-inflammatory properties of minocycline have been identified as an important feature of the action of this drug in various models of ischemic stroke [14,33,34,35]. Therefore, in the present study, we investigated the effect of minocycline on selected proteins implicated in the inflammatory response after ischemic stroke, including TNFα, HSP70, and the RNA-binding protein HuR.

TNFα is produced by various cell types, including neurons, microglia, astrocytes, and endothelial cells [28,36]. Previous studies have indicated that the prompt induction of TNFα mRNA in the photothrombotic model likely originates from ischemic neurons and not from activated glial cells [37]. In addition, TNFα is one of the earliest cytokines to appear in the context of the inflammatory reaction after ischemic brain injury, contributing to the stimulation of the inflammatory process in the cerebrospinal fluid and blood serum [38,39]. In contrast to TNFα protein, HSP70 (Heat Shock Protein 70) plays a protective role during the inflammatory process by inhibiting the response of pro-inflammatory cytokines such as TNFα and interleukin-1 [40]. Moreover, published research has shown that HSP70 regulates inflammation both intracellularly, where it exhibits anti-inflammatory properties, and extracellularly, where it may enhance immune responses [41,42,43,44]. Within this context, RNA-binding proteins (RBP) of the ELAV (embryonic lethal abnormal visual) family, which include the ubiquitously expressed HuR protein (also known as ELAV1) and three neuron-specific members (HuB, HuC, and HuD), may also play an important role. These RBPs act post-transcriptionally and influence the post-synthesis fate of the target transcripts, thereby regulating gene expression by binding to distinct signatures (known as ARE, adenine-uracil-rich elements) [45]. HuR, the most extensively studied member, is primarily localized in the nucleus and is known to affect the stability, translation, pre-mRNA splicing, and nuclear export of target mRNAs [42,45,46,47]. Within the general context of inflammation, HuR can play a dual role by participating in both non-inflammatory and inflammatory pathways. For example, HuR can bind to HSP70 mRNA in both in the nuclear and cytoplasmic compartments [48,49]. Under conditions of oxidative stress [50], it can promote a defense response in vulnerable neurons, thus directly blocking or delaying neuronal death [46,51]. On the other hand, HuR protein may also promote inflammation by interacting with mRNAs that encode pro-inflammatory cytokines such as TNFα [45,52,53,54,55]. Considering that both HSP70 and TNF-α transcripts are targeted by HuR, it was of interest to examine the role of this RBP following FCI and explore the effect of minocycline on its protein expression.

Finally, given the importance of understanding the crosstalk between molecular parameters and motor functions following FCI, motor tests were also carried out in this work (Figure 1). Specifically, our goal was to assess whether minocycline can modulate inflammation and whether possible improvement at the molecular level translates into a better motor performance after non-reperfusion ischemic stroke. This outcome holds considerable importance and relevance in the pursuit of novel and more effective therapies for clinical intervention.

## 2. Results

In total, six rats died immediately after anesthesia administration, resulting in a mortality rate of 6.25% for the entire study. No complications were observed at the sites of inflammation and necrosis within the skin incision or at dye injection sites.

### 2.1. Characterization of the Peri-Infarct Area following Ischemic Stroke

Nissl staining was performed to localize the necrotic and the penumbra areas after ischemic stroke in rats in the ipsilateral hemisphere. This staining allowed an easy localization of the ischemic area, which appeared much brighter compared to the healthy tissues. In order to compare the dynamics of the necrosis and the peri-infarct tissue changes among all groups (n = 5), the following parameters were measured: brain area (the entire area of the slice), necrosis area in relation to the brain area, necrosis width, and necrosis depth. For each rat, three measurements were made for each variable, and the mean was calculated. Then, variable statistic was performed for the “brain area” variable (ANOVA, *p* = 0.838). This result confirmed the lack of statistical significance of the “brain area” variable, regardless of the administration of minocycline and the time interval. For the control group (n = 6) statistical calculations were also made for the “brain area” variable (ANOVA, *p* = 0.091).

Infarct formation was already well advanced at 12 h (S1 and S1 + m groups) after the induction of the photothrombotic stroke, as evidenced by the increase in the volume of the “necrosis area to brain area” (Figure 2b). The infarct volume reached its maximum after 24 h (S2 and S2 + m groups); then, the infarction volume decreased significantly on the second day (S3 and S3 + m groups). Statistical analysis revealed significant differences (the Mann Whitney U test, *p* = 0.008) in all tested variables, including “necrosis area to brain area”, “necrosis width”, and “necrosis depth” at each time interval between the groups without administration of minocycline (S1, S2 and S3) and with administration of minocycline (S1 + m, S2 + m and S3 + m). The greatest decrease in infarct volume was observed between the S2 and S2 + m groups, where the administration of minocycline resulted in an average decrease of 0.8% in the volume of the infarcted area compared to the group without the administration of minocycline (Figure 2b).

### 2.2. Effect of Minocycline on Inflammation in the Peri-Infarct Area following Ischemic Stroke

Immunohistochemical staining was used to characterize the morphological and biochemical responses in the peri-infarct area immediately adjacent to the necrosis. By labeling the neurons with the NeuN neuronal marker, we were able to determine the extent of the necrosis after stroke induction. Only living neurons were labeled with NeuN, while dead neurons were not. The images showed that the necrotic area had already formed 12 h after the stroke, reaching its maximum extension at 24 h (S2 group).e NeuN labeling was retained at the edge of the infarct focus and in the healthy tissue (see Appendix A). Next, we studied the expression of HuR, TNFα, and HSP70 proteins in living neurons by performing double immunolabelling in brain sections (HuR+NeuN, TNFα+NeuN, HSP70+NeuN) and analyzing five regions of interest (ROIs), as shown in Figure 2a. Concerning HuR protein, the variables “nNeuN”, “nHuR” and “relative HuR” were examined for the Ipsi1, Ipsi2, Ipsi3, Cont4, and Cont5 regions in all the experimental groups. The Control group showed the same level of “relative HuR”, approximately 71%, in five ROIs (Figure 3a). The “relative HuR” variable was similar in the S1 and S1 + m groups, decreased slightly in the S2 + m group, and decreased drastically in the S2 group (reaching a level similar to the Control group at 24 h). At 48 h post-stroke induction, the expression of “relative Hurt” in the S3 + m group did not fall below 82% (Figure 3a). A comparison between the groups without the administration of minocycline and the groups with minocycline at each time point showed a statistically significant difference for all the three variables in favor of the groups with minocycline (the Mann-Whitney U test, *p* = 0.002; Figure 3a). With respect to the TNFα protein, the variables “nNeuN”, “nTNFα”, and “relative TNFα” were examined for the Ipsi1, Ipsi2, Ipsi3, Cont4, and Cont5 regions in all experimental groups. The Cont4 and Cont5 regions did not show any signal in any of the experimental group. The Control group exhibited a very low level of “relative TNFα”, approximately 0.02%, in all ROIs (Figure 3b). A comparison between the groups without minocycline versus those with minocycline at each time point (the Mann-Whitney *U* test, *p* = 0.002) showed a more significant increase in TNFα expression in the groups without minocycline (Figure 3b). Concerning the HSP70 protein, the variables “nNeuN”, “nHSP70” and “relative HSP70” were examined for the Ipsi1, Ipsi2, Ipsi3, Cont4, and Cont5 regions. There were no HSP70-positive cells in the Cont4 and Cont5 regions in any experimental group (Figure 3c). The results revealed statistically significant differences between the groups without the administration of minocycline and those with minocycline at all time points examined for all the “nNeuN”, “nHSP70”, and “relative HSP70” variables (Mann-Whitney U test, *p* = 0.002). HSP70 expression in the minocycline groups was statistically significantly higher than in the groups without the administration of minocycline. The “relative HSP70” variable peaked 24 h after stroke induction in both groups (Figure 3c).

The effect minocycline on the amount of HuR, TNFα, and HSP70 proteins in peri-infarct and infarct tissues was evaluated using the Western blotting technique. Concerning HuR protein, there was a significantly higher HuR protein expression in rats from the S3 + m group compared to the S3 group after FCI induction. There were no statistically significant differences for the other groups (Mann-Whitney U test, *p* = 0.041; Figure 4a). With respect to the HSP70 protein, there was a statistically significant difference in the level of HSP70 protein expression between the S3 and S3 + m groups. There were no statistically significant differences in the other groups (Mann-Whitney U test, *p* = 0.025; Figure 4b). With respect to the TNFα protein, a non-parametric analysis was performed. After inducing FCI in the rats, the level of TNFα protein was statistically significantly higher in the S3 group than in the S3 + m group (Kruskall-Wallis test, *p* = 0.004; Dunn’s post hoc test, *p* = 0.0362). There were no statistically significant differences in the other groups (Figure 4c).

### 2.3. Effect of Minocycline on the Level of Motor Performance in Rats after FCI

In our study, we also aimed to assess whether the administration of minocycline can improve motor performance in rats after the induction of ischemic stroke. For this purpose, all the stroke animals underwent motor tests including the Horizontal Runway Elevated test (HRE), Grip Strength test (GST) and CatWalk™ XT. None of the animals had any health problems or complications following the procedures, and no animals were excluded from the tests, except for groups S1 and S1 + m, which had a short duration between general anesthesia and euthanasia, making it impractical to perform motor tests [17]. Using the GST test, we were able to analyze the strength of the front paws in stroke rats. Two variables were measured: the “peak pull force”, which represents the maximum strength of the animal, and the “time of peak pull force”, which represents average time in which the animal reached the maximum strength of the front paws. The variables at each time point were treated as multiple measures. Some of the variables did not have a normal distribution within the groups (Shapiro-Wilk test, *p* > 0.05); therefore, an ANOVA test was used [56,57]. The analysis of variance for the “peak pull force” variable for the groups showed a statistically significant difference between the S2 and S2 + m groups (ANOVA, *p* < 0.026). The results revealed no differences in the mean strength of animals among the studied groups before the procedure (which indicates a homogeneous starting point) and a negative effect of stroke induction. In each time point at which the measurement was taken was different from the others (Bonferroni *post hoc* test, *p* < 0.001; Figure 5a). For the “time of peak pull force” variable, the analysis was statistically significant (ANOVA, *p* < 0.00001). Each time peak at which the maximum power was reached was different from the other (Bonferroni *post hoc* test, *p* < 0.001). The animals from the group without minocycline lost some strength after FCI induction already on the first day. They also had lower levels of strength in their forelimbs. Therefore, they reached their peak strength within a very short time compared to the animals treated with minocycline. (Figure 5b).

The Horizontal Runway Elevated (HRE) test was performed to assess the effect of minocycline on motor performance in rats after FCI. Specifically, this test allowed us to evaluate the walking speed and the accuracy of beam crossing. To analyze changes in the motility of animals, all the experimental groups were compared based on the following two variables: “sum of errors” (number of paw slips) and “mean time of passage” through the beam. The comparison was carried out from the day before the induction of ischemic stroke (D6) until the day of euthanasia (DX). After four days of training (D2–D5), the rats always traversed the beam flawlessly on D5 (Figure 6a). In addition, during the HRE test, a dynamic learning process was observed in rats, with the acquisition of flawless motor skills on the beam followed by a rapid decline in motor skills after FCI induction. Following stroke, the administration of minocycline resulted in a reduction in the number of errors made by rats at both 24 h and 48 h post-stroke induction compared to those not receiving minocycline. Additionally, both the time elapsed since stroke induction and the use of minocycline had a statistically significant effect on the number of errors made by rats in the HRE test (ANOVA, *p* = 0.0309; Figure 6a). For the variable “mean time of passage”, the analysis revealed that this variable was affected by training (D2-D5), the time elapsed since FCI induction, and the administration of minocycline (ANOVA, *p* < 0.05; Figure 6b). The animals that did not receive minocycline had a higher mean time of passage than the animals treated with minocycline. Moreover, the administration of minocycline resulted in a statistically significant reduction in the time needed to cross the beam in rats both 24 h and 48 h following FCI (Bonferroni *post hoc* test, *p* < 0.05; Figure 6b).

In order to assess the effect of minocycline on various gait parameters, CatWalk™ XT system (CWT) was used. CWT allowed us to assess the effect of ischemic stroke on locomotor functions and detect possible improvements induced by minocycline administration. The gait efficiency was assessed based on the nine most commonly used parameters: “Base of Support”, “Print area”, “Intensity”, “Duty cycle”, “Max area of contact”, “Stride length”, “Support time”, “Swing speed”, and the “Regularity Index”. Given that the stroke focus was on the left motor cortex of the brain, gait analysis was only performed on the front right paw (RF) and hind right paw (RH). The “Swing Speed” variable allowed us to measure the speed of paw in the air. For this variable, the analysis of variance showed a statistically significant decrease in the RF and RH movement speed in rats after FCI induction. The animals treated with minocycline had a statistically significantly faster swing rate in air than the untreated animals at each time point (Figure 7a,b). For the “Stride length” variable, the analysis of variance revealed a statistically significant effect of both stroke induction and minocycline administration on the RF stride length. Following stroke, the animals not receiving minocycline made an approximately 1.5 cm shorter step compared to those treated with minocycline 48 h after FCI. The analysis of variance for the RH showed a statistically significant effect of both stroke induction and minocycline on the RH stride length. The post-stroke animals without minocycline had an approximately 3.0 cm shorter RF stride at 48 h compared to animals treated with minocycline (Figure 7c,d).

With respect to the “Regular Index” variable, the analysis of variance showed a statistically significant effect of minocycline on the level of step regularity. Animals without minocycline after the FCI challenge had a decrease in step regularity compared to animals treated with minocycline (Figure 8a). The “Duty cycle” (%) variable represents the ratio of stand time to step cycle (Duty cycle = stand time/step cycle). Stand time (s) was calculated by the duration of contact with the walkway of a specific paw. Step cycle (s) was calculated by the duration of two consecutive initial contacts of a specific paw (step cycle = stand time + swing time). Regarding this variable, the analysis of variance for the RF showed a statistically significant effect of stroke induction, substance, and time interval. Similarly, for the RH, the ANOVA demonstrated a statistically significant effect of stroke induction, substance, and time interval. Animals without minocycline had a higher “Duty cycle” ratio than animals receiving minocycline (Figure 8b,c). Concerning the other variables “Base of Support”, “Print paw area”, “Intensity of paw contact”, “Max area of paw contact”, and “ Support time” the analysis of variance revealed no statistically significant differences.

## 3. Discussion

Overall, the results obtained in our research show that early administration of minocycline after ischemic stroke inhibits the enlargement of the necrotic area. At the molecular level, this event is associated with an increase in the content of both HSP70 protein and the RNA-binding protein HuR, and a decrease in the amount of TNFα in the peri-infarct tissue. Importantly, minocycline administration also produced an improvement in certain motor parameters following FCI.

Specifically, using Nissl staining, we assessed the volume, width, and depth of the necrotic focus. In our study, the stroke focus covered 5.5–7.0% of the brain volume, which is similar to that observed in humans [19]. In our study, the stroke focus covered 5.5–7.0% of the brain volume, which is similar to that observed in humans. This similarity enhances the translational potential of the results obtained using this animal model for the development of new therapies for humans. The volume of the stroke focus in rats progressed significantly 12 h after the induction of ischemic stroke and reached its maximum volume after 24 h, followed by a decrease after 48 h. This dynamic pattern of changes in the area affected by ischemic stroke aligns with other studies [14,58]. In the groups that received a single dose of minocycline, we observed statistically significant reductions in the infarct volume, width, and depth of the necrotic focus, with particularly remarkable improvement found in 12-week-old male rats. Reduced infarct volume after the administration of minocycline has also been observed in other studies using various animal models of ischemic stroke (i.e., FCI and middle cerebral artery occlusion (MCAO)) [14,31,33,59,60,61,62].

Accumulating evidence indicates that inflammation may play a key role in the pathogenesis of stroke, making it an interesting target for therapeutic interventions. Ischemic stroke is accompanied by inflammation, which is associated with elevated levels of pro-inflammatory cytokines, chemokines, and peripheral blood leukocytes (e.g., neutrophils, monocytes, T lymphocytes) observed in these areas [14,28,63,64,65,66,67,68]. The development of inflammation and the infiltration of pro-inflammatory cells (e.g., monocytes) are considered to be one of the phenomena responsible for increasing the area of secondary damage [69]. Notably, studies on animal models of ischemic stroke have demonstrated that minocycline has some anti-inflammatory properties [35,61,62,70,71] that inhibit the expression of pro-inflammatory cytokines and reduce brain damage [72,73]. Within this context, several investigations have documented that neuronal ischemia increases the expression of TNFα as early as 1 h after the onset of ischemia in the MCAO model [74,75]. Furthermore, Kondo et al. and Meng et al. demonstrated that the expression of TNFα in cells other than neurons (i.e., microglia and astrocytes) becomes visible more than 24 h after the onset of a stroke [37,76]. Considering that inflammatory processes stimulate neuroprotective effects if they persist for a short time but cause neurodegenerative processes if they persist for a longer time [28], we also examined the time course of TNFα expression in neurons located in the peri-stroke tissue. Using double labeling with TNFα and the neuronal marker NeuN, we were observed TNFα expression in neurons as early as 12 h after the induction of ischemic stroke. Similarly, Li et al. and Yang et al. showed that within 6–12 h from the onset of symptoms, an increasing amount of TNFα was detected in the blood of human patients and in the cerebral tissues of rats following stroke [77,78]. We used Western blotting to assess the amount of TNFα protein in the peri-stroke tissue, which supported the immunohistochemistry results. The data analysis showed a statistically significant decrease in TNFα levels in the groups receiving minocycline. Consistent with these findings, other authors have also noted a decrease in TNFα content after the administration of minocycline in animal models of ischemic stroke [33,71,79]. In particular, Yang et al. showed that a single dose of minocycline given immediately after the start of reperfusion significantly inhibits microglial activation after 48 h, indicating that early administration of minocycline after stroke can reduce inflammation levels [79]. Therefore, our results may be related to the inhibitory effect of minocycline on the inflammatory response in the acute phase of stroke. As mentioned earlier, HSP70 protein regulates inflammation both intracellularly, where it appears to play an anti-inflammatory role, and extracellularly, where it may enhance immune responses. We found that under homeostatic conditions, as also reported in other studies [80,81], HSP70 levels are low. Using NeuN-HSP70 double labeling, we observed changes in the number of HSP70-positive neurons in the peri-infarct tissue as early as 12 h after stroke induction. Most importantly, HSP70 expression was statistically significantly higher in the groups receiving minocycline compared to the groups without minocycline. The Western blotting data supported the results of immunohistology, demonstrating a statistically significant difference in HSP70 protein expression on the second day after stroke induction between the minocycline-treated groups and the untreated animals. Consistently, other authors have also reported an increase in the amount of HSP70 after ischemic stroke [80,81,82,83], suggesting that minocycline itself is likely responsible for the increase in HSP70 protein levels in the peri-stroke tissue [84,85].

Given that both HSP70 and TNF-α transcripts are targeted by HuR [45,50], we also determined the level of HuR protein in the peri-stroke tissue in order to examine the direct effect of minocycline on this RNA-stabilizing protein. Using NeuN-HuR double immunolabeling, we conducted analyses comparing the groups without minocycline to the groups with minocycline, which revealed a statistically significant increase in the number of neurons in the minocycline-treated groups. Furthermore, in the group without minocycline administration, the number of HuR-positive cells already decreased dramatically 24 h after the stroke, indicating the developing of inflammation in the peri-stroke tissue. Western blotting data analysis confirmed a statistically significantly higher HuR protein expression in the minocycline-treated group compared to the untreated group 48 h after FCI induction. These results suggest that following FCI, this RNA-binding protein promotes a protective response by shifting its binding towards HSP70 instead of TNFα. Notably, at 48 h, we observed a concomitant increase in HSP70 protein content and a decrease in TNFα expression. Similarly, Jamison et al. demonstrated a correlation between HuR and HSP70 protein levels following ischemic stroke, highlighting the functional importance of the correlation between HuR and the concomitant appearance of HSP70 in the same neurons [51]. Hence, our results suggest that HuR protein may play a role in modulating inflammation mitigate some secondary damage in the aftermath of stroke. Given that low-level inflammation has a protective effect on neurons while chronic high-level inflammation is detrimental [86,87,88,89], determining the exact role of HuR protein in the inflammation process may contribute to the improvement of new post-stroke therapies, especially in the context of neuronal protection.

In the present study, we also assess whether the positive biochemical and morphological changes observed under the influence of minocycline would translate into improved motor functions. In this regard, we would like to emphasize that our improved and minimally invasive animal model of ischemic stroke allows for motor tests to be performed after FCI [17]. The data from selected tests showed positive changes in gait, balance level, speed of movement, and leg strength. Specifically, the results obtained from the GST indicate that minocycline-treated animals achieved a statistically significant higher level of maximum paw strength duration and reached the maximum strength later compared to the untreated animals at each time point. In this respect, Soliman et al. also used GST in the MCAO model of ischemic stroke and demonstrated a statistically significant improvement in the animals treated with minocycline [31]. Using the HRE test, we examined the effect of minocycline on the speed of movement along the beam and the number of errors during the transition in stroke rats. The animals that did not receive minocycline were characterized by a statistically significantly higher number of errors (paw slipping off the beam during the transition) both 24 h and 48 h after FCI induction. We also document that, after FCI, the administration of minocycline resulted in a statistically significant reduction in the time needed by rats to cross the beam at each time point. In this regard, Soliman et al. along with Li et al., using the HRE test in the MCAO model, also demonstrated that the administration of minocycline improved the motor function of animals [31,90]. Using the CWT, we explored whether there was an improvement in gait parameters in the animals receiving minocycline after FCI. For the following gait parameters: stride length, duty cycle, swing speed, and regularity index, we observed statistically significant improvements in favor of minocycline administration. The animals receiving minocycline had a statistically significantly faster swing rate, longer strides, increased step regularity, and lower standing time to step cycle ratios compared to the untreated animals. All the motor tests used allowed us to observe even very subtle improvements in many parameters of gait, balance, and fitness in the acute phase of ischemic stroke in the rats treated with minocycline. It is worth emphasizing that these tests are much more accurate than neurological scales and provide objective changes in the parameters. In conclusion, our results indicate that motor tests are a valuable tool to verify whether positive changes, both biochemical and morphological, translate into improved motor efficiency after ischemic stroke. Improved motor performance is one of the goals of therapy after ischemic stroke in humans. Therefore, the use of these approaches can strongly improve the translation of therapies from animal models to patients with ischemic stroke [91,92,93].

## 4. Materials and Methods

### 4.1. Animals

Experimental protocols were approved by the Local Bioethics Committee at the Medical University of Silesia, Katowice, Poland, and were consistent with international guidelines on the ethical use of animals. All rats were bred in the Department for Experimental Medicine, Medical University of Silesia in Katowice, Poland. Throughout the entire study, the animals were housed in a temperature-controlled and humidity-controlled room with a 12 h light/dark cycle. They had ad libitum access to water and standard rat chow. From the 5th week of the animals’ life until the start of the study, routine behaviors were implemented according to our protocol (see our previous publication [17]), resulting in a significant stress reduction in the animals.

Ninety-six male Long-Evans (LE) rats weighing 223–239 g were used. The rats were randomly divided into six experimental groups (Figure 1). Groups S1 (n = 14), S2 (n = 14), and S3 (n = 14) consisted of animals without minocycline administration, and the time between ischemic stroke induction and euthanasia was 12 h, 24 h, and 48 h, respectively. Groups S1 + m (n = 14), S2 + m (n = 14), and S3 + m (n = 14) consisted of animals with minocycline administration, and the time elapsed between ischemic stroke induction and euthanasia (DX) was equal to 12 h, 24 h, and 48 h, respectively. The Control group (n = 12) used in this study consisted of animals that did not undergo any procedures. For this animal model of ischemic stroke, in accordance with the 3R principles and our previous results, only one control group was created [17]. In order to verify motor deficits and the relative effects of minocycline, all groups underwent the Horizontal Runway Elevated test, CatWalk™ XT, and Grip Strength test both before the induction of ischemic stroke and on the euthanasia day.

### 4.2. Animal Model

A full and detailed description of the ischemic model used in the present study is available in our previous publication [17]. During the stroke induction surgery, the animals were placed under general anesthesia induced by intraperitoneal administration of xylazine hydrochloride (10 mg/kg of body weight) and ketamine hydrochloride (100 mg/kg of body weight). The infarct was created in the posterior motor cortex (the spot was marked with the coordinates 0.5 mm anterior to bregma and 3.0 mm laterally from the centerline). The center was determined using an optical fiber with a 5 mm diameter. A non-transparent mask was used to protect the remaining skull areas from the laser light. The skull was irradiated with white light at a wavelength of 560 nm and 3200 K [94] (KL2500, LCD SCHOTT, Mainz, Germany). The irradiation lasted for 15 min. The dye injection took place during the first minute of irradiation. Specifically, a Bengal Rose (BR) solution (20 mg of BR in 1 mL of PBS (Sigma-Aldrich, St. Louis, MO, USA)) was slowly injected at a dose of 1 mL/kg of body weight through a pre-inserted polyethylene catheter into the tail vein (*lateral caudal vein*) of each rat. After the irradiation was finished, the optical fiber was removed and the incisions were sutured with skin sutures.

### 4.3. Treatment with Minocycline

Rats in groups S1 + m, S2 + m, and S3 + m received a single dose of minocycline (Sigma-Aldrich, St. Louis, MO, USA) at 1 mg/kg body weight, dissolved in 1 mL 0.9% NaCl, administered intravenously 10 min after the ischemic stroke induction. The chosen single dose [32,79,95] of minocycline was based on previous studies that reported neuroprotection in the ischemic stroke model under similar conditions. Based on the reports obtained after a single dose in our experience, we decided to administer the lowest possible dose of 1 mg/kg body weight. This allows for the design of further in-depth studies, with the possibility of modulating the dose volume or changing the frequency of minocycline administration to the animals. Animals in groups S1, S2, and S3 received the same volume of saline intravenously.

### 4.4. Horizontal Runway Elevated Test

In this study, motor deficits resulting from the induction of ischemic stroke in rats were tested using the horizontal runway elevated (HRE) test. The HRE test is commonly used in animal models to evaluate forelimb and hindlimb function and coordination [96,97,98,99]. For each experimental group (n = 12–14), two variables were measured: the sum of the average number of errors made and the average time taken to pass through the slat. The HRE test consisted of an elevated ladder apparatus and a camera (GoPro Hero 8, San Mateo, CA, USA). The camera was necessary to register the animals’ passage over the beam. The recorded videos were needed for a quantitative and qualitative analysis of the rats’ gait. The HRE test used in this study was constructed by our team [17]. Successful completion of the HRE test required six runs for each animal: three runs from cage A to cage B, and three runs from cage B to cage A. The rats began by crossing the slat from cage A to cage B. This action was repeated in the opposite direction until the sixth successful passage. The final run concluded in cage A for all rats. The HRE test was divided into two phases: the training phase (D2–D5) and the official passage phase (D6–DX). After the training period, during the official passage phase, all rats performed one official pre-stroke passage on the stroke induction day (D6), and one official passage post-stroke on the day of euthanasia (DX).

### 4.5. Grip Strength Test

The forelimb Grip Strength test (GST) was performed using an electronic digital force gauge grip-strength meter (47200, UGO Basil, Gemonio, VA, Italy). The use of the GST following ischemic stroke is consistent with previous publications [100,101,102,103]. For each experimental group (n = 12–14), two variables were measured: the peak force exerted by the animal while gripping the sensor bar and the time of peak force. The back part of the rat’s body was gently held and pulled back until its front paw loosened its grip on the sensor, and the maximum grip strength was automatically recorded. GST was performed three times on the day of ischemic stroke induction (D6) and three times after the ischemic stroke induction, on the euthanasia day (DX). The duration of the GST was 5 s; if the rat did not pull/hold the sensor within that time, the measurement was not recorded, and the animal was excluded.

### 4.6. CatWalk™ XT

An automated quantitative gait analysis system was used to assess rat motor function and coordination: CatWalk™ XT (CWT; Noldus, Wageningen, The Netherlands). Each test group consisted of the same number of rats (n = 10). The CWT system has a unique application in the study of strokes [104,105]. The CWT assessment was conducted in a quiet and darkened room. When rats made contact with the glass plate, the light signals from their paws were reflected and transformed into digital messages by a video camera. The performance of the CWT was divided into two phases: the training phase (5 days long; D1–D5) and the official passage phase (2 days long; D6–DX). After the training days, during the official passage phase, all rats performed one official pre-stroke passage on the stroke induction day (D6) and one official passage post-stroke on the day of euthanasia (DX). The five-day training phase helped to minimize stress in rats and ensure the performance of three consecutive uninterrupted runs [90,106]. A correctly performed CWT assessment consisted of three undisturbed runs for each rat [90]. A series of gait statistics were automatically generated when the system identified and marked each footprint. The statistics included the following parameters: “Base of Support”, “Print paw area”, “Intensity of paw contact”, “Max area of paw contact”, “Support time”, “Duty cycle”,” Stride length”, “Swing speed”, and “Regularity Index” [90,106,107,108,109].

### 4.7. Histological Analysis

Histological evaluation required the collection of brain tissues from each experimental group (n = 6) and the Control group (n = 6) in accordance with the assigned time interval between the induction of ischemic stroke and euthanasia. The rats were anesthetized and transcardially perfused with 4% paraformaldehyde. The fixed brains were then removed and postfixed in a paraformaldehyde solution overnight at 4 °C. Subsequently, the brains were dehydrated, embedded in paraffin, and sectioned using a microtome (Leica Microsystems, Mannheim, Germany) into coronal planes (−2.50 mm to −2.90 mm from bregma) with 7 µm-thick slices. The distance between each of the 10 sections used per animal was 50 µm.

#### 4.7.1. Nissl Staining

The preparations were stained with 1 g/L cresyl violet dye (Sigma-Aldrich, St. Louis, MO, USA) in water, following the classical Nissl staining protocol with our own modifications [17,110]. Images of the peri-infarct cortex area in the ipsilateral hemisphere and a reference calibration slide were acquired using a digital camera (Olympus OM-D E-M10 Mark IV, Tokyo, Japan). The area of the infarct in each section, identified by pale staining, was measured using the ImageJ 1.43 software (Madison, WI, USA) [111]. The same software was also used to calculate the infarct volume, width, and depth of each brain.

#### 4.7.2. Immunohistochemistry

Immunohistochemistry assay was used to measure the levels of HuR, TNFα, and HSP70 proteins expression in paraffin-embedded brain tissue slides. The tissue slices were blocked with 0.1% Triton X-100 (Sigma-Aldrich) and 10% serum (goat normal serum, Vector Labs). The sections were then incubated overnight at 4 °C with primary antibodies against HuR, NeuN, TNFα, and HSP70 (Table 1). The primary antibodies were followed by the following fluorochrome-conjugated secondary antibodies: goat anti-mouse Alexa Fluor^®^ 488 (green Alexa; Cambridge, UK) and goat anti-rabbit Alexa Fluor^®^ 594 (red Alexa; Cambridge, UK), which were incubated for 1h at room temperature. Finally, the sections were mounted on slides using a DAPI-containing medium. To initially characterize the changes in the brain tissue after FCI in ipsilateral hemisphere, images were obtained from double-immunolabeled sections of HuR-NeuN, TNFα-NeuN and HSP70-NeuN. Five regions of interest (ROIs) were incorporated, as shown in Figure 2a. These included ROIs at the lateral and medial edges of the infarct and more distant sites in the contralateral cerebral cortex. The immuno-positive cells were counted in the field of view from the ROI areas: Ipsi1, Ipsi2, Ipsi3, Cont4, and Cont5. The variables “nNeuN”, “nX”, and “relative X” (n = number of immunohistochemically positive cells in the field of view; X—double HuR+NeuN, TNFα+NeuN, or HSP70+NeuN immuno-positive cells from the tested protein in the field of view) were calculated. We also calculated the relative value from the growth: “relative X”(%) = nX/(nX + nNeuN). The sections of double-immunolabeled ROIs were captured and analyzed under a fluorescence microscope (Olympus BX43, Tokyo, Japan) using cellSens Standard software (Olympus) and processed with ImageJ 1.43usoftware. The ROIs with size of 500 × 500 μm were analyzed.

### 4.8. Western Blotting

Samples were collected from all the experimental groups (n = 6) and the Control group (n = 5) to determine proteins expression using the Western blot technique. The rats were sacrificed by decapitation and their brains were immediately removed from the skulls. The area containing the infarct, along with approximately 1.0 mm of the surrounding tissue, was dissected for analysis. For each rat, the material was collected using a biopsy punch of the same size. The tissue samples were immediately frozen on dry ice and stored in a freezer at −80 °C. Protein levels were measured using the Bradford method. Bovine albumin was used as an internal standard. The proteins were diluted in 2x SDS protein gel loading solution, boiled for 5 min, and separated on a 12% SDS-PAGE gel. HuR, TNFα and HSP70 antibodies were according to the instructions provided in each datasheet (Table 1). The nitrocellulose membrane signals were detected using chemiluminescence. The same membranes were re-probed with an α-tubulin antibody and used to normalize the data. Densitometric values obtained with the ImageJ image-processing program were subjected to statistical analysis for the Western blot data.

### 4.9. Statistical Analysis

The statistical analysis was performed using Statistica 13.1 software (Dell, Austin, TX, USA). The significance level was set at *p* = 0.05. Descriptive statistics of the studied variables in groups were presented, and tests examining the normality of distribution were carried out when appropriate (Shapiro-Wilk test *p* > 0.05). Additionally, *Levene*’*s* test was performed to assess the homogeneity of variance. Based on these results, parametric (ANOVA) or non-parametric tests (Kruskal-Wallis, *U* Mann-Whitney) were selected, and when statistically significant results were obtained, appropriate *post hoc* tests (Bonferroni’s and Dunn’s) were carried out. The values are presented in charts as the mean ± standard deviations for variables analyzed with parametric tests, and as medians along with the quartile range (IQR) for variables analyzed with non-parametric tests.

## 5. Conclusions

Severe cerebral ischemia leads to neuronal damage and death in the ischemic area. Several previous studies have suggested that minocycline protects neurons from damage caused by cerebral ischemia. Our study confirmed that the administration of minocycline increases the viability of neurons and reduces neurodegeneration caused by ischemia. The effects of minocycline led to a significant reduction in the infarct volume after ischemia. Minocycline also affected selected parameters of inflammation: it reduced the content of TNFα in the peristroke tissue, while increasing the levels of HSP70 and HuR proteins in the same area. After ischemic stroke, minocycline could play an important role in modulating inflammation, particularly at the level the stability and/or translation of target mRNAs. The results show that the minocycline influences the level of ubiquitously expressed HuR protein (RNA-binding protein). The increase in the level of HuR protein translates into the improvement of molecular and motor parameters after ischemic stroke. Therefore, the role of the HuR protein in modulating inflammation may be important, although further in-depth research is required. Motor tests have demonstrated that the reduction in inflammation parameters and the decrease in the area of brain tissue damage after the administration of minocycline directly translates into a better motor performance. Therapies based on targeting inflammation appear to be a very interesting and effective approach in post-ischemic stroke therapy available to most patients. We believe that understanding both the cellular and molecular changes in the brain throughout all phases of ischemic stroke is essential for developing the most effective therapies capable of counteracting the negative effects of this devastating process.

## Figures and Tables

**Figure 1 ijms-24-09446-f001:**
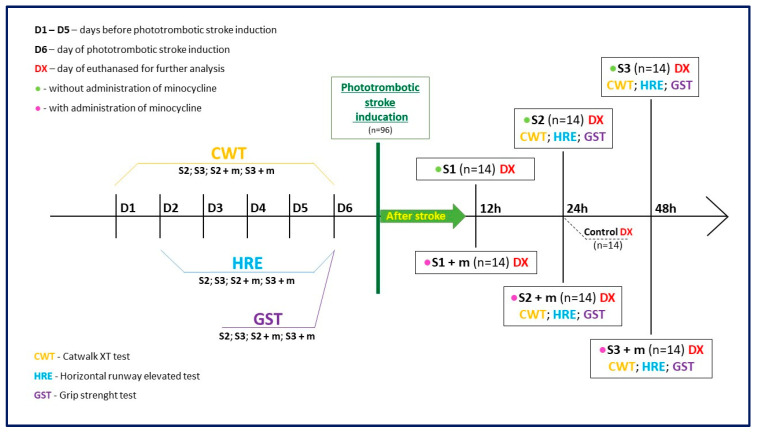
Graphical flowchart providing an overview of the study. Groups S1, S2, and S3 consisted of animals without minocycline administration; the time elapsed between ischemic stroke induction and euthanasia was equal to 12 h, 24 h, and 48 h, respectively. Groups S1 + m, S2 + m, S3 + m included animals with minocycline administration; the time elapsed between ischemic stroke induction and euthanasia (DX) was equal to 12 h, 24 h, and 48 h, respectively. In order to verify the effect of minocycline administration on the level of motility after stroke induction, the following motor tests were also performed on the animals: CatWalk™ XT (CWT), Grip Strength test (GST), and the Horizontal Runway Elevated (HRE) test.

**Figure 2 ijms-24-09446-f002:**
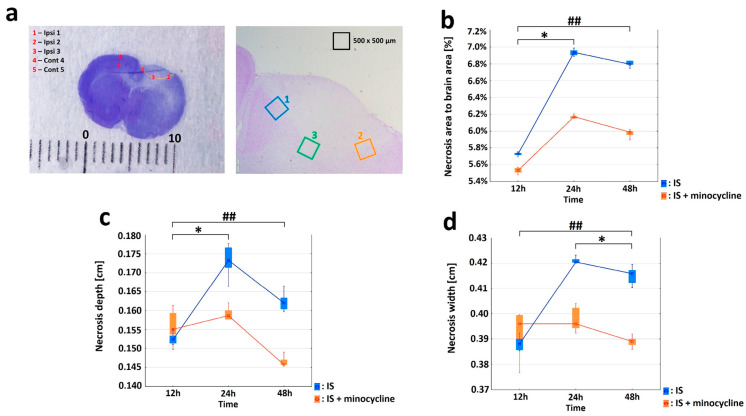
Effect of minocycline on infarct volume. Comparison of dynamics of volume, width, and depth of the necrotic area after ischemic stroke induction in rats: Nissl staining. Administration of minocycline caused less intense histopathological changes compared to the groups without minocycline (IS: ischemic stroke groups, IS + minocycline: ischemic stroke + minocycline groups). (**a**) A Nissl-stained coronal section of a rat brain 24 h after the induction of a photothrombotic stroke showing a clearly defined infarct area (left panel) and the ROIs (right panel). The scale is in millimeters (Ipsi: ipsilaterally hemisphere, Cont: contralateral hemisphere). (**b**) Changes in the average necrosis area (%) over time and the relative effect of minocycline. The results for the variable “necrosis area to brain area” for groups without minocycline (Kruskall-Wallis test, *p* = 0.005) and groups with minocycline (Kruskall-Wallis test, *p* = 0.002). For the pairs S1 + m vs. S2 + m and S1 *vs*. S2, a statistically significant difference was demonstrated (Dunn’s *post hoc* test, *p* < 0.05), where * *p* < 0.05 for intragroup comparisons. For each time interval, the differences between the studied groups were significant (Mann-Whitney U test, *p* = 0.008), where ## *p* < 0.01 for the comparison of groups without versus groups with the administration of minocycline. (**c**) The necrosis width spread after stroke induction. Within the groups: with minocycline (Kruskall-Wallis test; *p* = 0.0652), groups without minocycline (the Kruskall-Wallis test, *p* = 0.004). For S1 vs. S2, a statistically significant difference was demonstrated (Dunn’s *post hoc* test, *p* < 0.05), where * *p* < 0.05 for intragroup comparisons. For each time interval, the differences between the studied groups were significant (Mann-Whitney *U* test, *p* = 0.008), where ## *p* < 0.01 for the comparison of groups without versus groups with the administration of minocycline. The analyses for groups S1 and S1 + m did not show any statistically significant differences. (**d**) The necrosis depth deepened after stroke induction. Within the groups: with minocycline (Kruskall-Wallis test, *p* = 0.008), without minocycline (Kruskall-Wallis test, *p* = 0.003). For the pairs S2 + m vs. S3 + m and S1 vs. S2, a statistically significant difference was demonstrated (Dunn’s *post hoc* test, *p* < 0.05), where * *p* < 0.05 for intragroup comparisons. For each time interval, the differences between the studied groups were significant (Mann-Whitney U test, *p* = 0.008), where ## *p* < 0.01 for the comparison of groups without versus groups with the administration of minocycline. The analysis between groups S1 and S1 + m did not show any statistically significant differences.

**Figure 3 ijms-24-09446-f003:**
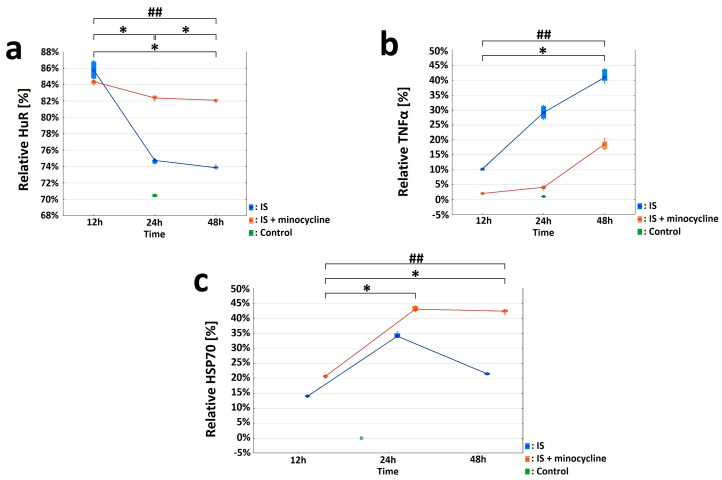
Effect of minocycline on selected parameters of inflammation after the induction of ischemic stroke in rats. HuR, TNFα, and HSP70 were investigated in the peri-infarct area after ischemic stroke induction: immunohistochemistry (IS: ischemic stroke groups; IS + minocycline: ischemic stroke + minocycline groups; Control: control group). (**a**) HuR and NeuN double - immunolabeled neurons. The variables “nNeuN”, “nHuR”, and “relative HuR” show statistically significant difference (KruskalWallis test, *p* = 0.0002). S1 vs. Control, S2 vs. Control, S3 vs. Control, S1 vs. S3, and S1 + m vs. S3 + m showed a statistically significant difference (Dunn’s *post hoc* test, *p* < 0.05). S1 + m vs. Control, S2 + m vs. Control, S3 + m vs. Control, S1 vs. S3, and S1 + m vs. S3 + m showed a statistically significant difference for the relative “HuR” variable (Dunn’s *post hoc* test, *p* < 0.05), where * *p* < 0.05 for intragroup comparisons and comparisons with the control group. Comparing the groups without the administration of minocycline and with the administration of minocycline at each time point revealed the presence of statistically significant differences for all the three examined variables, “nNeuN”, “nHuR”, and “relative HuR”, in favor of the minocycline groups (Mann-Whitney *U* test, *p* = 0.002), where ## *p* < 0.01 for the comparison of groups without versus groups with the administration of minocycline. (**b**) TNFα and NeuN double-immunolabeled neurons. The obtained results for the “nNeuN”, “nTNFα”, and “relative TNFα” variables showed statistically significant differences for all variables only between the S2 vs. Control groups (Kruskall-Wallis, *p* = 0.0005; Dunn’s *post hoc* test, *p* < 0.05). For each studied variable, the pairs S1 vs. S3 and S1 + m *vs*. S3 + m were statistically significant (Dunn’s *post hoc* test, *p* < 0.05), where * *p* < 0.05 for intragroup comparisons and comparisons with the control group. The occurrence of differences between the groups without minocycline and with minocycline at each time point were compared for the “nNeuN”, “nTNFα”, and “relative TNFα” variables. All three variables were statistically significantly different at each time point between the groups without minocycline and with minocycline (Mann-Whitney U test, *p* = 0.002), where ## *p* < 0.01 for the comparison of groups without versus groups with the administration of minocycline. (**c**) HSP70 and NeuN double-immunolabeled neurons. The values of the “nNeuN”, “nHSP70”, and “relative HSP70” variables showed a statistically significant difference between all groups without minocycline and those with minocycline (Kruskall-Wallis test, *p* = 0.0001). The pairs S1 vs. S3 and S1 + m vs. S3 + m showed a statistically significant difference (Dunn’s *post hoc* test, *p* < 0.05) for the “nNeuN” and “nHSP70” variables. The pairs S1 vs. S3, S1 + m vs. S2 + m, and S1 + m vs. S3 + m showed a statistically significant difference in the “relative HSP70” variable (Dunn’s *post hoc* test, *p* < 0.05), where * *p* < 0.05 for intragroup comparisons and comparisons with the control group. The “nNeuN”, “nHSP70”, and “relative HSP70” variables results showed statistically significant differences (Mann-Whitney *U* test, *p* = 0.002) between the groups without minocycline and those with minocycline at all time points examined, where ## *p* < 0.01 for the comparison of groups without versus groups with the administration of minocycline.

**Figure 4 ijms-24-09446-f004:**
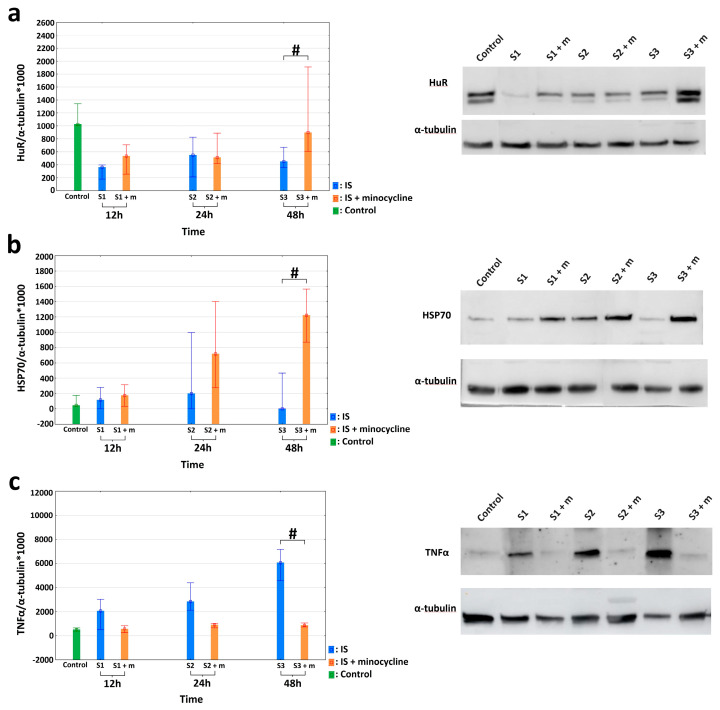
Effects of minocycline administration on selected markers of inflammation after photothrombotic stroke in rats. HuR, TNFα, and HSP70 levels in the peri-infarct area after ischemic stroke induction; Western blotting (IS: ischemic stroke groups; IS + minocycline: ischemic stroke + minocycline groups; Control: control group). (**a**) HuR protein levels: the results confirmed a significantly higher HuR protein expression in the S3 + m rats compared to the S3 group after FCI induction (the Mann Whitney U, *p* = 0.041), where # *p* < 0.05 for the comparison of groups without versus groups with the administration of minocycline. (**b**) HSP70 protein levels: the results confirmed a statistically significant difference in HSP70 protein expression between the S3 and S3 + m groups (Mann Whitney U, *p* = 0.025), where # *p* < 0.05 for the comparison of groups without versus groups with the administration of minocycline. (**c**) TNFα protein levels: there was a statistically significant difference between the S3 and S3 + m groups (Kruskall-Wallis test, *p* = 0.004; Dunn’s *post hoc* test, *p* = 0.0362), where # *p* < 0.05 for the comparison of groups without versus groups with the administration of minocycline.

**Figure 5 ijms-24-09446-f005:**
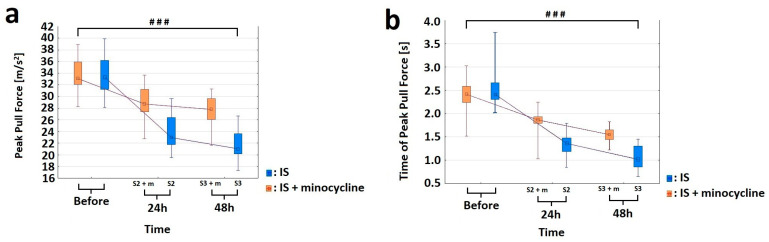
Effect of minocycline on the strength level of the front paws after prothrombotic stroke induction in rats: Grip Strength test. Administration of minocycline after stroke increased front paw strength relative to animals without minocycline. (**a**) For the “peak pull force” variable, the analysis of variance showed statistically significant differences in the mean values of the variable between the S2 vs. S2 + m groups (ANOVA, *p* = 0.0254) and as a function of time (ANOVA, *p* = 0.0000). The interaction between these two factors, group and time, was significant (ANOVA, *p* = 0.0000). The comparison showed that each time point at which the measurement was made differed from the other (The Bonferroni *post hoc* test, *p* < 0.001), where ### *p* < 0.001 for comparison of groups without versus groups with the administration of minocycline. (**b**) For the “time of peak pull force” variable, the analysis was significant for the two factors “group” (ANOVA, *p* = 0.00001) and “time” (ANOVA, *p* = 0.0000), as well as for the interaction between these two factors (ANOVA, *p* = 0.00000). The comparison showed that each peak time at which the maximum power was reached was different from the other (Bonferroni *post hoc* test, *p* < 0.001), where ### *p* < 0.001 for the comparison of groups without versus groups with the administration of minocycline.

**Figure 6 ijms-24-09446-f006:**
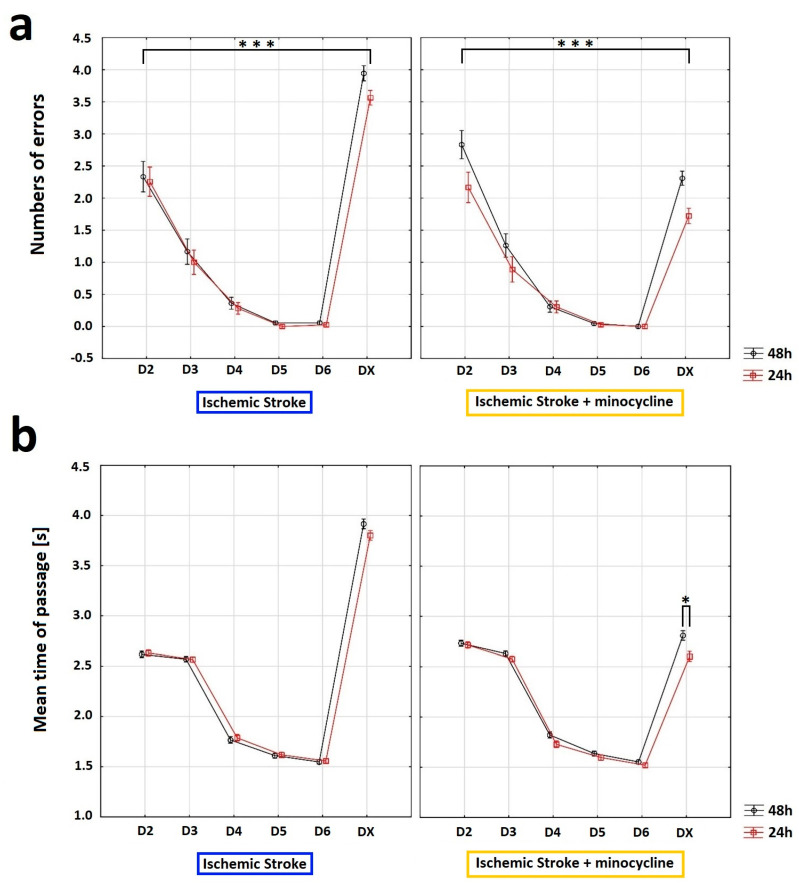
Influence of minocycline on the speed of passage and the number of errors after induction of prothrombotic stroke in rats: Horizontal runway elevated test (HRE). The administration of minocycline after stroke increased the speed of passage through the beam and reduced the number of errors made. (**a**) Concerning the “sum of errors”, an analysis of variance was performed: “beam passage time” (ANOVA, *p* = 0.0009) and “minocycline” (ANOVA, *p* = 0.009) influenced the number of errors made by the rats. Moreover, the interaction between “minocycline” and “beam passage time” showed a statistically significant result (ANOVA, *p* = 0.0000). The “beam passage time” interaction with groups (pre-stroke vs. 24 h post FCI vs. 48 h post FCI) also reached a statistically significant result (ANOVA, *p* = 0.0309). Training days (D2–D5) and testing days (D6 and DX) differed significantly in terms of the average number of errors (ANOVA, *p* = 0.0000). Interactions between “minocycline” over time were statistically significant (ANOVA, *p* = 0.0000), as well as the beam passage time interaction between groups (pre-stroke vs. 24 h post FCI vs. 48 h post FCI; Bonferroni *post hoc* test, *p* < 0.05), where *** *p* < 0.001 for intragroup comparisons. (**b**) With respect to the “mean time of passage”, the analysis of “mean time of passage” was performed. The variable was affected by training (D2–D5), time elapsed since FCI induction (24 h post-induction or 48 h post-induction), and minocycline administration (ANOVA, *p* < 0.05). The animals without minocycline had a higher mean time of passage. In the case of the animals with minocycline, time played a role—the animals passed the beam slower 48 h after the procedure than after 24 h (Bonferroni *post hoc* test, *p* < 0.05). The administration of minocycline resulted in a statistically significant reduction in the time needed for minocyclinetreated rats to pass the beam both 24 h and 48 h after FCI compared to the animals without minocycline (the Bonferroni *post hoc* test, *p* < 0.05); where * *p* < 0.05 for intragroup comparisons.

**Figure 7 ijms-24-09446-f007:**
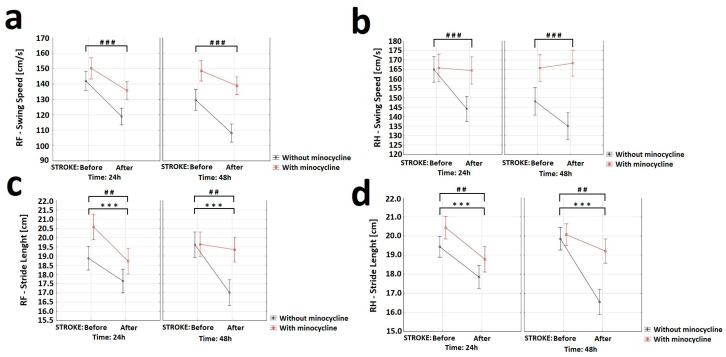
Effect of minocycline on gait speed and stride length in rats after ischemic stroke: CatWalk XT. Selected gait parameters improved after minocycline administration for both the right front (RF) and right hind (RH) paws. For the variable “Swing speed” (**a**,**b**) the analysis showed a decrease in RF (ANOVA, *p* = 0.0003) and RH (ANOVA, *p* = 0.001) movement speed after FCI induction. The animals receiving minocycline had a statistically significantly faster swing speed than animals without minocycline in each time group (ANOVA, *p* = 0.0001). Concerning the “Stride length” variable (**c**,**d**), the analysis revealed the effect of both stroke induction (ANOVA, *p* = 0.003) and minocycline (ANOVA, *p* = 0.008) on the RF stride length. Following FCI, animals without minocycline made an approximately 1.5 cm shorter step compared to animals with minocycline 48 h following FCI. The analysis of variance for RH showed the effect of both stroke induction (ANOVA, *p* = 0.0000) and minocycline (ANOVA, *p* = 0.009) on RH stride length. Following FCI, the animals without minocycline had an approximately 3.0 cm shorter RF stride compared to the animals with minocycline 48 h following FCI, where *** *p* < 0.001 for intragroup comparisons and ## *p* < 0.01, ### *p* < 0.001 for the comparison of groups with and without the administration of minocycline.

**Figure 8 ijms-24-09446-f008:**
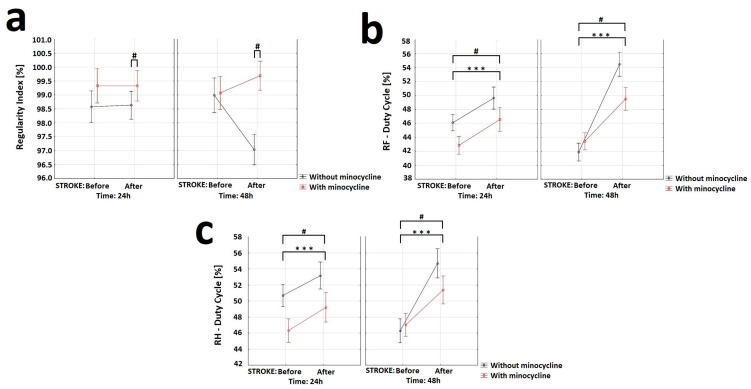
Effect of minocycline on gait regularity and step interval in rats after ischemic stroke induction: CatWalk XT. Selected gait parameters improved after minocycline administration for both the right front (RF) and right hind (RH) paws. Regarding the “Regular Index” variable (**a**), the analysis showed the effect of minocycline on the level of steps regularity (ANOVA, *p* = 0.013). Groups S2 and S3 displayed a decrease in step regularity after FCI compared to animals given minocycline (S2 + m and S3 + m). With respect to the “Duty cycle” variable (**b**,**c**), the analysis of the RF showed the effect of stroke induction (ANOVA, *p* = 0.00000), minocycline (ANOVA, *p* = 0.0307), and time interval (ANOVA, *p* = 0.006) on this parameter. For the RH, the results showed the effect of stroke induction (ANOVA, *p* = 0.000057), minocycline (ANOVA, *p* = 0.03589), and time interval (ANOVA, *p* = 0.00751) on this parameter, where *** *p* < 0.001 for intragroup comparisons, and # *p* < 0.05 for the comparison of groups with and without administration of minocycline.

**Table 1 ijms-24-09446-t001:** Table of the antibodies used in the experiments.

Antibody	Abbreviation	Type	Company	Catalog Number	Dilution	Order
HuR/ELAV1	HuR	mouse monoclonal	Santa Cruz Biotechnology	sc-5261	for IHCP we used 1:1000	primary antibodies
Neuronal Marker	NeuN	rabbit monoclonal	Abcam	ab177487	for IHCP we used 1:1000	primary antibodies
Anti-TNFα	TNFα	mouse monoclonal	Santa Cruz Biotechnology	sc-52746	for IHCP we used 1:1000	primary antibodies
Anti-Hsp70	HSP70	mouse monoclonal	Abcam	ab2787S	for IHCP we used 1:500	primary antibodies
HuR/ELAV1	HuR	mouse monoclonal	Santa Cruz Biotechnology	sc-5261	for WB we used 1:1000	primary antibodies
Anti-TNFα	TNFα	rabbit monoclonal	Abcam	ab205587	for WB we uded 1:500	primary antibodies
HSC70/HSP70	HSP70	mouse monoclonal	Santa Cruz Biotechnology	sc-24	for WB we uded 1:500	primary antibodies
Anti-α-Tubulin	α-Tubulin	mouse monoclonal	Sigma-Aldrich	T0198	for WB we udes 1:1000	primary antibodies
Goat anti-mouse Alexa Fluor^®^ 488	green Alexa	goat anti-mouse IgG	Abcam	ab150113	for IHCP we used 1:500	secondary antibodies
Goat anti-rabbit Alexa Fluor^®^ 594	red Alexa	goat anti-rabbit IgG	Abcam	ab150080	for IHCP we used 1:500	secondary antibodies
Goat anti-mouse IgG-HRP	anti mouse	goat anti-mouse IgG	Sigma-Aldrich	A4416	for WB we used 1:3000	secondary antibodies
Goat anti-rabbit IgG-HRP	anti rabbit	goat anti-rabbit IgG	Merck Millipore	AP156P	for WB we used 1:3000	secondary antibodies

## Data Availability

The dataset used and/or analyzed during this study is available from the corresponding author upon reasonable request.

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
