# Peer review of "After Ischemic Stroke, Minocycline Promotes a Protective Response in Neurons via the RNA-Binding Protein HuR, with a Positive Impact on Motor Performance"

_ijms, 2023, doi:10.3390/ijms24119446_

Round 1

Reviewer 1 Report

Overall, this is an exciting study in which the authors investigated the beneficial effects of minocycline after ischemic stroke.

Major comment:

We acknowledge the authors’ expertise in inducing focal cerebral ischemic stroke. However, it is essential to demonstrate comparable cerebral blood flow reductions in both stroke groups, with and without treatment. Therefore, to provide a detailed description of the methodology employed to assess blood flow following focal cerebral ischemic stroke and include the corresponding results of cerebral blood flow measurements within the manuscript. We believe that this addition will strengthen the manuscript. Thank you for considering this suggestion.

Author Response

Dear Reviewer

Thank you for your time and valuable comments, which will allow us to design and perform experiments in animal models even better in the future.

In our experiment, we chose a focal ischemic photostroke (FCI) model induced by clotting dye (Bengal Rose) after white light activation. Low invasive and very precise model of ischemic stroke. We described all the advantages of this model in detail in our previous publication (PMCID: PMC9776070 and DOI: 10.3390/brainsci12121671). This FCI model is a very effective model because a permanent cut-off of blood flow in the vessels of the irradiated area. In addition, the specificity of this model allows you to eliminate the natural recanalization of the vessel by dissolving the thrombus, because this Bengal Rose is resistant to the action of the substance dissolving the thrombus.

In this experience, we wanted to focus on those types of ischemic strokes that are no longer amenable to treatment with the most effective techniques such as alteplase or thrombectomy to restore blood flow. The latest reports show that this type of the most effective treatment can still be used in less than 10% of all patients suffering from ischemic stroke. The rat FCI model mimics situations where circulation/blood flow can no longer be restored to an ischemic area. And the most important goal of therapy in this case is to stop the growing necrosis and restore balance in the penumbra zone. Therefore, in our experiment we did not focus on the aspect of the influence of minocycline on changes in blood flow in vessels affected by ischemia. That is the reason why we did not monitor this flow in the ischemic zone.

Using histological techniques in the FCI model, we are able to very precisely define the area of necrosis and penumbra. Monitoring the dynamics of changes in these areas and the impact of the test substance on these areas was our main goal in this experiment. In addition, with precise and permanent ischemia of the motor cortex, we had full control over motor deficits in rats and reflected the dynamics of changes in the examined areas during motor tests. These tests allowed us to verify the effectiveness of the tested substance (minocycline) also at the motor level changes.

Yours faithfully

Katarzyna Pawletko 

Reviewer 2 Report

The authors investigated minocycline treatment neuron protective effect after ischemic stroke and underlying pathway. They found that minocycline increases neurons’ viability, reduces the neurodegeneration, infract volume and TNFа expression, increases HSP70 and HuR protein expression. So, the authors concluded that minocycline promotes a protective response in neurons via the RNA-binding protein HuR with a positive impact on motor performance after ischemic stroke.

Overall, this study is clearly described, study design is rational and the data are evidently presented. This paper contributes to the research community of anti-inflammation treatment in ischemic stroke field. However, I still have the following comments in order to improve the manuscript.

1.      This study’s novelty is not enough since minocycline’s neuroprotection after ischemic stroke has been widely investigated (e.g., PMID:31756420; PMID:19807907). The authors should emphasize the difference between this study and other similar studies in the introduction.

2.      In “4.3. Treatment with minocycline” paragraph of “4. Materials and Methods” section, the authors explained why minocycline’s dosage (1mg/kg) was chosen, however, references cited in here does not support this dosage (1mg/kg). Minocycline dosage is 6mg/kg from reference 95, and Minocycline dosage is 3mg/kg from reference 32 and 79. Please clarify this point.

3.      The authors did not explain why minocycline was injected on 10 minutes after ischemic model induction and should provide evidence to support it.

4.      It will be helpful if the authors can quantify the double immunolabelling cells (HuR+NeuN; TNFа+NeuN; HSP70+NeuN) in the Figure 1S from the supplementary materials and see if there have any difference between the two groups.

Author Response

Dear Reviewer

Thank you for your time and valuable comments. We took all comments into consideration. They will allow us to improve the manuscript and aspects of future research.

In relation to comments, we also send our answers

1. In our experience, we focused on those types of strokes that no longer qualify for the most effective treatment to restore blood flow (only about 10% of all strokes qualify for treatment with alteplase or thrombectomy). The situations of patients who do not qualify for the most effective ischemic stroke therapies are perfectly imitated by the FCI model of focal cerebral ischemia. We included all the advantages and our improvements of the FCI model in the previous publication (PMCID: PMC9776070 and DOI: 10.3390/brainsci12121671). We have also included a reference to this article in the methodology section.

Although the results in the cited publications suggest similar conclusions about the effects of minocycline, the model in the PMID:31756420 study is bilateral occlusion of the common carotids (BCCAo) for 25 min and subsequent reperfusion. Also in the publication PMID:19807907, another model was used - middle cerebral artery occlusion (MCAo).

Compared to the ischemic stroke models cited above, the FCI model allows for more reproducible results, does not allow for the possibility of reperfusion, and more closely mimics what happens in humans (i.e. the ischemic area occupies a relatively small area of the entire cerebral cortex, represents the situation of 90% of all stroke that does not qualify for reperfusion). In addition, FCI model is characterized by a low burden for animals and allows to verify the effect of the test substance using motor tests, where it is not always possible after MCAo (a very large area is affected by ischemia, preventing the animals from moving efficiently).

We initially outlined the reasons for our choice of the FCI model and the need to investigate the effect of minocycline on this particular model in lines 53 to 61.

2. We sincerely apologize for this citation error, as the referenced papers [32, 79,95] deal with the administration of a single dose, not its size of dose. This has already been corrected in the manuscript.

In our experience, we decided on the lowest effective dose (1 mg/kg b.w) of minocycline, based on the results and experience of scientists from our Department (publication with hemorrhagic strokes with the use of minocycline PMCID: PMC9899762). Promising results encouraged us to test a single dose of 1mg/kg b.w. in a model of ischemic stroke in rats.

3.  The dose of minocycline was given only after the induction of ischemic stroke so that the whole experience in the animal model could resemble real situations in patients. People affected by ischemic stroke also receive treatment only after the onset of symptoms of ischemic stroke.

For the research to be of a basic character, which was very important to us. The results of the planting obtained in this way can be used to design further in-depth studies in order to increase the effectiveness of the action by, for example, increasing the dose, increasing the frequency of dose administration as well as the time of administration of the drug from the onset of the stroke. Therefore, in this experiment, it was 10 minutes from the end of irradiation of the rat skull, because this is the time needed to suture the scalp, dress the wounds and secure the animal after all procedures.

4. Immunopositive cells were counted in the field of view from ROIs: Ipsi1, Ipsi2, Ipsi3 and Cont4 and Cont5. Each ROI was 500 × 500 μm in size. Variables "nNeuN", "nX" and "relative X" where (n= number of immunohistochemical positive cells in the field of view; X- double HuR+NeuN; TNFα+NeuN; HSP70+NeuN immunopositive cells from the tested proteins in the field of view). With these two values, we could also calculate the relative value from the formula: "relative X" (%) = nX / (nX+nNeuN). For a description of the counting method, see section 4.7.2 Immunohistochemistry.

In Figure 1S, we have included two exemplary groups (2S and 2S+M) and photos in additional materials, so as not to increase the size of the main text. In the main text we have included the most important statistical results resulting from our calculations. For all three proteins tested (HuR, TNFalpha and HSP70), the quantification statistics are shown in Figure number 3.

Yours faithfully Katarzyna Pawletko 

Round 2

Reviewer 1 Report

We appreciate the authors' response—no further comments from the reviewer.